# Verrucosidin Derivatives from the Deep Sea Cold-Seep-Derived Fungus *Penicillium polonicum* CS-252

**DOI:** 10.3390/ijms23105567

**Published:** 2022-05-16

**Authors:** Yanhe Li, Xiaoming Li, Xin Li, Suiqun Yang, Bingui Wang, Honglei Li

**Affiliations:** 1CAS and Shandong Province Key Laboratory of Experimental Marine Biology, Institute of Oceanology, Chinese Academy of Sciences, Nanhai Road 7, Qingdao 266071, China; liyanhe1025@163.com (Y.L.); lixmqdio@126.com (X.L.); lixin871014@163.com (X.L.); suiqunyang@163.com (S.Y.); 2Laboratory of Marine Biology and Biotechnology, Qingdao National Laboratory for Marine Science and Technology, Wenhai Road 1, Qingdao 266237, China; 3College of Marine Science, University of Chinese Academy of Sciences, Yuquan Road 19A, Beijing 100049, China; 4Center for Ocean Mega-Science, Chinese Academy of Sciences, Nanhai Road 7, Qingdao 266071, China

**Keywords:** deep-sea-derived fungus, *Penicillium polonicum*, verrucosidin derivatives, structure elucidation, antimicrobial activity

## Abstract

Six novel verrucosidin derivatives, namely, poloncosidins A–F (**1**–**6**), together with one known analogue (**7**), were isolated and identified from the deep-sea-derived fungus *Penicillium polonicum* CS-252, which was obtained from cold-seep sediments collected in the South China Sea at a depth of 1183 m. Their structures were mainly established on the basis of a detailed interpretation of NMR spectroscopic and mass spectrometric data. The relative and absolute configurations of compounds **1**–**6** were determined by ECD calculations and a DP4+ probability analysis. Compounds **1**–**5** represent the first examples of verrucosidins with a 2,5-dihydrofuran ring which is uncommon among the known analogues. These compounds exhibited inhibitory activities against several human and aquatic pathogens with MIC values ranging from 4 to 32 μg/mL.

## 1. Introduction

Verrucosidins belong to a family of highly reduced polyketides, generally sharing a methylated α-pyrone, a conjugated polyene linker, and an epoxidated tetrahydrofuran ring [1]. All verrucosidin derivatives previously isolated and characterized from the fungal genus Penicillium were found to possess bioactivities such as antibacterial and GRP78/BiP down-regulator activities [2,3]. *Penicillium polonicum* is a ubiquitous fungus found as saprophyte in diverse environments [4], from which bioactive molecules with diverse chemical structures have been reported including diketopiperazine, α-pyrone derivative, dimeric anthraquinone, and taxol [5,6,7,8,9,10]. The verrucosidin biosynthesis gene cluster was also reported in *P. polonicum*, which confirmed the ability of the fungus to produce related secondary metabolites [11,12].

In our ongoing research for the discovery of bioactive secondary metabolites from marine-derived fungi [13,14,15], the fungal strain *P. polonicum* CS-252, which was obtained from sediments collected at the deep-sea cold seep area in the South China Sea, attracted our attention due to its unique HPLC profiles which showed peaks with different UV absorptions from those in our HPLC database. A chemical investigation of the culture extract of *P. polonicum* CS-252 led to the characterizations of six new verrucosidin derivatives, designated as poloncosidins A–F (**1**–**6**, Figure 1), together with a known analogue, deoxyverrucosidin (**7**). Among them, compounds **1**–**5** represent the first examples of verrucosidins with a 2,5-dihydrofuran ring. Details of the isolation, purification, structure elucidation, and biological evaluation of compounds **1**–**7** are provided herein.

## 2. Results and Discussion

The fungus *P. polonicum* CS-252 was cultured on a rice solid medium for chemical investigation. The obtained culture was extracted thoroughly with EtOAc to produce an organic extract that was further fractionated and purified with a combination of column chromatography over Lobar LiChroprep RP-18, Si gel, and Sephadex LH-20 as well as by prep. TLC to yield compounds **1**–**7**.

Poloncosidin A (**1**) was obtained as a colorless oil and its molecular formula was assigned as C_24_H_32_O_4_ on the basis of HRESIMS data, corresponding to nine indices of hydrogen deficiency. The ^1^H-NMR and HSQC spectra of **1** (Table 1 and Appendix A) revealed the signals of nine methyls with one doublet (*δ*_H_ 1.16, *J* = 6.4 Hz) eight singlets (*δ*_H_ 1.33, 1.66, 1.88, 1.93, 1.95, 1.96, 2.00, and 3.82), four singlet olefinic protons (*δ*_H_ 5.53, 5.61, 5.93, and 6.16), and a quartet proton at *δ*_H_ 4.62 (1H, q, *J* = 6.4 Hz). The ^13^C-NMR and DEPT data (Table 2 and Appendix A) indicated all 24 carbon resonances including nine methyls, five methines (with four olefinic at *δ*_C_ 127.2, 137.1, 137.2, and 139.1 as well as an oxygenated at *δ*_C_ 81.6), and ten nonprotonated carbons (with eight olefinic, one carbonyl, and one oxygenated). The aforementioned spectroscopic data suggested the characteristics of a verrucosidin derivative, which are similar to those of deoxyverrucosidin (**7**) previously described in *Penicillium* sp. in 2005, with the difference that the epoxide moiety in **7** was replaced by a double bond in **1**. This difference was supported by the NMR evidence that signals for an oxygenated methine (*δ*_H_ 3.45; *δ*_C_ 67.5, CH-13) and an oxygenated nonprotonated carbon (*δ*_C_ 67.4, C-14) in the NMR spectra of deoxyverrucosidin (**7**) [3] were not present in those of **1**, while resonances for an olefinic methine (*δ*_H_ 5.61, H-13; *δ*_C_ 127.2, C-13) and an olefinic nonprotonated carbon (*δ*_C_ 138.2, C-14) were observed in the NMR spectra of **1** (Table 1 and Table 2). These data indicated that the epoxidated tetrahydrofuran ring in deoxyverrucosidin was replaced by a 2,5-dihydrofuran ring in **1**. This deduction was supported by the key HMBC from H-11 (*δ*_H_ 5.53) and H_3_-21 (*δ*_H_ 1.33) to C-13 (*δ*_C_ 127.2) and from H_3_-22 (*δ*_H_ 1.66) to C-13, C-14 (*δ*_C_ 138.2), and C-15 (*δ*_C_ 81.6) (Figure 2). The planar structure of **1** was thus identified as the first example of verrucosidins possessing a 2,5-dihydrofuran moiety.

The relative configuration of compound **1** was established by an analysis of its NOESY data (Figure 3). Key NOESY enhancement from H-15 to H_3_-21 indicated the cofacial orientation of these protons, which established the relative configuration of the 2,5-dihydrofuran moiety. The geometries of the double bonds at C-6, C-8, and C-10 were determined to be *E* configurations by the NOESY correlations from H-7 to H-9 and H_3_-17 and from H-9 to H-11 (Figure 3). To establish the absolute configuration of **1**, the time-dependent density functional (TDDFT)-ECD calculations at the BH&HLYP/TZVP level were performed in Gaussian 09. The calculated ECD spectrum for the (12*S*,15*R*)-isomer of compound **1** matched well with the experimental curve, leading to the establishment of the absolute configuration of **1** as 12*S*, 15*R* (Figure 4).

Poloncosidin B (**2**) was obtained as a colorless oil and the molecular formula was assigned as C_23_H_30_O_4_ based on an analysis of the HRESIMS data with the same hydrogen deficiency index as that of **1**. A comprehensive analysis of the ^1^H- and ^13^C-NMR data of **2** indicated a similar structure to that of **1**, only differing in the absence of a substituent. Signals for a methyl (*δ*_H_ 1.93; *δ*_C_ 9.9, CH_3_-16) and an olefinic nonprotonated carbon (*δ*_C_ 108.6, C-2) in the NMR spectra of **1** were found to be absent in those of **2** (Table 1 and Table 2). In contrast, compound **2** exhibited an additional resonance at *δ*_H_ 5.61 (1H, s, H-2) in the ^1^H-NMR spectrum that correlates with an olefinic methine carbon at *δ*_C_ 87.9 (C-2), accounting for the shielding effect of the carbonyl group at C-1. The key HMBC correlations from H-2 to C-3 (*δ*_C_ 170.7) and C-4 (*δ*_C_ 106.0) as well as other key HMBC and COSY correlations (Figure 2) established the structure of compound **2** as shown.

The relative configuration of the 2,5-dihydrofuran ring and the geometries of the double bonds at C-6, C-8, and C-10 were determined to be the same as those of compound **1** with an analysis of the NOESY correlations (Figure 3). The absolute configuration of **2** was assigned as 12*S* and 15*R*, which was evidenced by a comparison of the overlaid experimental and calculated ECD curves at the BH&HLYP/TZVP level (Figure 5).

The molecular formula of poloncosidin C (**3**) was determined as C_21_H_28_O_4_ according to the HRESIMS ion peaks at *m*/*z* 345.2057 [M + H]^+^ (calcd for C_21_H_29_O_4_, 345.2060) and 367.1875 [M + Na]^+^ (calcd for C_21_H_28_O_4_Na, 367.1880), required eight indices of hydrogen deficiency. Detailed interpretation of the ^1^H- and ^13^C-NMR data revealed **3** to be structurally related to compounds **1** and **2**, featuring a methylated *α*-pyrone moiety and a 2,5-dihydrofuran ring linked with a shorter polyene linker. This deduction was supported by the NMR evidence that resonances for one of the trisubstituted double bonds in the conjugated polyene linker, comprising a singlet methyl (*δ*_H_ 1.95; *δ*_C_ 17.9, CH_3_-19), a singlet olefinic methine (*δ*_H_ 5.93; *δ*_C_ 137.1, CH-9), and an olefinic nonprotonated carbon (*δ*_C_ 131.3, C-8) in the NMR spectra of **1** disappeared in those of **3**. Combined with the key HMBC correlations from H-7 (*δ*_H_ 6.08) to C-5 (*δ*_C_ 158.4) and C-9 (*δ*_C_ 138.3) as well as from H_3_-18 (*δ*_H_ 1.33) to C-9 (*δ*_C_ 138.3), C-10 (*δ*_C_ 87.3) and C-11 (*δ*_C_ 127.1) (Figure 2), the planar structure of compound **3** was thus established.

The observed NOESY correlations between H-13 and H_3_-18 helped to establish the relative configuration of the 2,5-dihydrofuran moiety (Figure 3). The *E*-geometry of the double bonds at C-6 and C-8 was determined by the observed NOEs from H-7 to H-9 and H_3_-15 (Figure 3). The absolute configuration of **3** was assigned as 10*S* and 13*R* based on the results the of TDDFT-ECD calculations at the BH&HLYP/TZVP level as shown in Figure 6.

Poloncosidin D (**4**) was isolated as a colorless oil. The molecular formula was established as C_21_H_30_O_4_ from its HRESIMS data with seven indices of hydrogen deficiency. The ^1^H- and ^13^C-NMR spectra of **4** revealed a 2,5-dihydrofuran moiety with three sets of conjugated double bonds, conforming to the characteristics of verrucosidin derivatives. However, additional resonances for two carbonyl carbons (*δ*_C_ 198.5, C-3 and 171.4, C-1) were observed in the ^13^C-NMR spectrum of **4** that are quite different from those of compounds **1**–**3**. The COSY correlation from H-2 (*δ*_H_ 4.52) to H_3_-14 (*δ*_H_ 1.21) as well as HMBC correlations from H_3_-14 to C-1 and C-3, from H-5 (*δ*_H_ 7.19) to C-3, C-7 (*δ*_C_ 140.8), and C-16 (*δ*_C_ 17.8), and from H_3_-21 (*δ*_H_ 3.60) to C-1 established the structure of **4**, which possesses the same polyketide skeleton as that of compound **3**, but without the formation of the *α*-pyrone moiety.

The relative configuration of **4** was established by the observed NOESY correlations (Figure 3), except for the stereogenic center of C-2. To establish the stereochemistry, TDDFT-ECD calculations for four possible isomers of **4** [**4a**-(2*R*,10*S*,13*R*), **4b**-(2*S*,10*S*,13*R*), **4c**-(2*R*,10*R*,13*S*), and **4d**-(2*S*,10*R*,13*S*)] (Figure 7) were performed, and then two isomers [**4a**-(2*R*,10*S*,13*R*) and **4b**-(2*S*,10*S*,13*R*)] were recommended for **4** (Figure 8), which were further clarified by the DP4+ probability analysis. Both the proton and carbon data of the two possible isomers were calculated based on the DP4+ protocol and the results were analyzed with the experimental values [16]. The statistical analysis indicated that the isomer [**4a**-(2*R*,10*S*,13*R*)] was the equivalent structure with a probability of 99.91% (Figure 8 and Appendix A). The absolute configuration of **4** was thus assigned.

The molecular formula of poloncosidin E (**5**) was assigned as C_24_H_32_O_4_ based on the accurate mass data measured by HRESIMS, which was identical to that of **1**. The ^1^H- and ^13^C-NMR data of **5** were virtually identical to those obtained for **1** (Table 1 and Table 2), except for the shielded location of C-5 at *δ*_C_ 155.9 and de-shielded location of C-18 at *δ*_C_ 22.7 in the ^13^C-NMR spectrum of **5**, conforming to the *γ*-gauche effect [17]. Moreover, resonances for C-6 (*δ*_C_ 125.6) and C-7 (*δ*_C_ 133.0) also moved to the shielded field in the ^13^C-NMR spectrum of **5**. A detailed analysis of the COSY and HMBC correlations of **5** (Figure 2) revealed that it possessed the same planar structure as **1**, but with *Z*-geometry for the double bond at C-6. The structure of compound **5** was thus established with the above NMR data.

Unfortunately, the NOESY spectrum of **5** was not obtained due to instability and low abundance, and compound **5** was actually isolated as a highly impure mixture. To further define the geometry of the double bond at C-6, a DP4+ probability analysis was performed. The steric configuration of the 2,5-dihydrofuran moiety was determined as the same as that of compounds **1**–**4** based on the NMR data comparisons and biosynthetic origins. Two possible isomers [**5a**-(6*E*,12*S*,15*R*) and **5b**-(6*Z*,12*S*,15*R*)] of compound **5** were thus generated for DP4+ calculations (Figure 9). The ^13^C-NMR data calculated for the isomer [**5b**-(6*Z*,12*S*,15*R*)] matched well with the experimental data of **5** (probability = 100%), which led to the assignment of the steric configuration for **5**.

Poloncosidin F (**6**) was obtained as a colorless oil with a molecular formula of C_23_H_32_O_6_ as determined by the HRESIMS data. Its ^1^H- and ^13^C-NMR data (Table 1 and Table 2) revealed a close structural relationship with deoxyverrucosidin (**7**) [3]. One of the visible differences were that the signals for a methoxy group 3-OCH_3_ (*δ*_H_ 3.83; *δ*_C_ 60.2) in the NMR spectra of deoxyverrucosidin (**7**) were absent in those of 6. Moreover, a de-shielded resonance for C-3 (*δ*_C_ 177.0) and a shielded field shift for C-2 (*δ*_C_ 89.9) were clearly observed in the ^13^C-NMR spectrum of 6 (Table 2), compared to those of **7** [3]. These spectroscopic features as well as in considering the molecular formula revealed that the ester linkage of the α-pyrone ring in deoxyverrucosidin (**7**) was hydrolyzed in 6. Supported by the HMBC correlations from H_3_-16 (*δ*_H_ 1.57) to C-1 (*δ*_C_ 165.3), C-2, and C-3 as well as from H_3_-17 (*δ*_C_ 1.72) to C-3, C-4 (*δ*_C_ 111.3), and C-5 (*δ*_C_ 155.4) (Figure 2), the planar structure of **6** was established.

The *E*-geometry of five conjugated double bonds at C-2, C-4, C-6, C-8, and C-10 were assigned by the NOE correlations from H_3_-17 to H_3_-16 and H_3_-18, from H_3_-18 to H_3_-19, and from H-9 to H-7 and H-11. NOE correlations from H-13 to H_3_-22 and H_3_-23 indicated the cofacial orientation of these groups, while the cross-peak from H-15 to H_3_-21 assigned these groups at the opposite face. Similarly, the absolute configuration of **6** was determined by the TDDFT-ECD calculations in Gaussian 09. The ECD spectrum calculated at the BH&HLYP/TZVP level for isomer (12*S*,13*S*,14*R*,15*R*)-**6** matched well with the experimental curve (Figure 10), leading to the establishment of the absolute configuration of **6**.

In addition to the new compounds **1**–**6**, the known analogue deoxyverrucosidin (**7**) was also isolated and identified by a detailed spectroscopic analysis and comparisons with the reported data [3].

The obtained compounds, except for compound **5**, were assayed for antimicrobial activities against human-, aqua-, and plant-pathogenic microbes. These compounds showed inhibitory activity against some of the tested bacterial strains (MIC ≤ 32 μg/mL, Table 3). Among them, compounds **4** and **6** displayed inhibitory activity against *Escherichia coli* EMBLC-1, *Klebsiella pneumoniae* EMBLC-3, *Pseudomonas aeruginosa* QDIO-4, *Vibrio alginolyticus* QDIO-5, and *V. parahemolyticus* QDIO-8 with MIC values ranging from 4 to 32 μg/mL. Compound **4** also inhibited methicillin-resistant *Staphylococcus aureus* (MRSA) EMBLC-4 with an MIC value of 16 μg/mL. Moreover, compound **1** showed activity against *P. aeruginosa* QDIO-4, *V. alginolyticus* QDIO-5, and *V. parahemolyticus* QDIO-8 with MICs 8.0, 8.0, and 4.0 μg/mL, respectively, whereas compound **7** was bactericidal against *P. aeruginosa* QDIO-4 and *V. parahemolyticus* QDIO-8 (each with an MIC value of 8.0 μg/mL). These data indicated that verrucosidin derivatives without the formation of the *α*-pyrone moiety exhibited a broad spectrum of antimicrobial activity against the tested strains (**4** and **6** vs. **1**–**3**, and **7**). In addition, the methylation at C-2 and the 2,5-dihydrofuran ring in these compounds enhanced their activities (**1** vs. **2** and **7**), while the absence of a trisubstituted double bond in the polyene linker (compound **3**) likely had an adverse effect on the antibacterial activity.

## 3. Experimental Section

### 3.1. General Experimental Procedures

The instruments, organic solvents, and reagents used in this work were the same as those used in our previous reports [13,14,15].

### 3.2. Fungal Material

The fungus *P. polonicum* CS-252 was isolated from the cold seep sediments, which were collected from South China Sea (depth 1183 m) in September 2020. The fungal strain was identified as *Penicillium polonicum* based on the ITS region sequence, which is the same (100%) as that of *P. polonicum* FNG3 (with accession no. MZ301257.1). The sequence data were deposited in GenBank with the accession no. OK175489.1. The strain is being preserved at the Key Laboratory of Experimental Marine Biology, Institute of Oceanology, Chinese Academy of Sciences (IOCAS).

### 3.3. Fermentation, Extraction and Isolation

For chemical investigations, 1 L Erlenmeyer flasks with a solid medium containing rice were autoclaved at 120 °C for 20 min before inoculation. Each flask contained 70 g rice, 0.3 g peptone, 0.5 g yeast extract, 0.2 g corn steep liquor, 0.1 g monosodium glutamate, and 100 mL of naturally sourced and filtered seawater obtained from the Huiquan Gulf of the Yellow Sea near the campus of IOCAS. The fresh mycelia of *P. polonicum* CS-252 were grown on a PDA medium at 28 °C for seven days, inoculated into the autoclaved medium and then statically cultured for 30 days at room temperature. The fermented rice substrate was fragmented mechanically and extracted thoroughly with EtOAc. The above EtOAc solutions were combined and concentrated under reduced pressure to produce 70.0 g organic extract.

The extract was subjected to vacuum liquid chromatography (VLC) over Si gel and fractionated using solvent mixtures of increasing polarity consisting of petroleum ether (PE) and EtOAc (20:1 to 1:1) and finally with CH_2_Cl_2_-MeOH (50:1 to 1:1) to yield 10 fractions (Frs. 1−10). Purification of Fr. 3 (4.1 g), performed with column chromatography (CC) over Lobar LiChroprep RP-18 with a MeOH-H_2_O gradient (from 1:9 to 10:0), yielded 10 subfractions (Frs. 3.1−3.10). Fr. 3.7 (113 mg) was purified with CC on Sephadex LH-20 (MeOH) to obtain compounds **4** (7.8 mg) and **6** (3.8 mg). Fr. 3.8 (215.0 mg) was subjected to CC on Si gel (CH_2_Cl_2_-MeOH, 200:1 to 150:1), and then purified by prep. TLC and CC on Sephadex LH-20 (MeOH) provided compounds **1** (52.4 mg), **2** (6.2 mg), and **3** (4.0 mg). Fr. 3.9 (75 mg) was purified with CC on Si gel (CH_2_Cl_2_-MeOH, 300:1), and then purified with CC on Sephadex LH-20 (MeOH) to obtain compound **7** (23.1 mg). Fr. 4 (10.4 g) was further fractionated with CC over Lobar LiChroprep RP-18 elution with a MeOH-H_2_O gradient (from 1:9 to 10:0) to yield 10 subfractions (Frs. 4.1−4.10). Fr. 4.3 (87 mg) was separated with CC on Si gel and Sephadex LH-20 (MeOH) to obtain compound **5** (7.4 mg).

*Poloncosidin A* (**1**) colorless oil; ${\left[\alpha \right]}_{D}^{25}$ = +65, c 0.10, MeOH; UV (MeOH) *λ*_max_ (log *ε*) 229 (2.64), 303 (2.23) nm; ECD (1.04 mM, MeOH) *λ*_max_ (Δ*ε*) 200 (–1.39), 223 (+1.39), 265 (+1.01), 321 (+3.56) nm; ^1^H- and ^13^C-NMR data, Table 1 and Table 2; HRESIMS *m*/*z* 385.2373 [M + H]^+^ (calcd for C_24_H_33_O_4_, 385.2373).

*Poloncosidin B* (**2**) colorless oil; ${\left[\alpha \right]}_{D}^{25}$ = +51, c 0.10, MeOH; UV (MeOH) *λ*_max_ (log *ε*) 230 (2.66), 304 (2.25) nm; ECD (1.10 mM, MeOH) *λ*_max_ (Δ*ε*) 221 (+1.03) nm, 261 (+0.32) nm, 308 (+5.77) nm; ^1^H- and ^13^C-NMR data, Table 1 and Table 2; HRESIMS *m*/*z* 371.2216 [M + H]^+^ (calcd for C_24_H_31_O_4_, 371.2217).

*Poloncosidin C* (**3**) colorless oil; ${\left[\alpha \right]}_{D}^{25}$ = +33, c 0.15, MeOH; UV (MeOH) *λ*_max_ (log ε) 246 (2.68), 315 (3.03) nm; ECD (1.16 mM, MeOH) *λ*_max_ (Δε) 201 (–1.86), 217 (+2.24), 250 (+1.09), 311 (+0.49) nm; ^1^H- and ^13^C-NMR data, Table 1 and Table 2; HRESIMS *m*/*z* 345.2057 [M + H]+ (calcd for C_21_H_29_O_4_, 345.2060), 367.1875 [M + Na]^+^ (calcd for C_21_H_28_O_4_Na, 367.1880).

*Poloncosidin D* (**4**) colorless oil; ${\left[\alpha \right]}_{D}^{25}$ = +38, c 0.14, MeOH; UV (MeOH) *λ*_max_ (log ε) 232 (2.63) nm, 299 (2.32) nm; ECD (1.16 mM, MeOH) *λ*_max_ (Δε) 200 (–2.93), 223 (+0.05), 251 (–0.72), 296 (+1.64) nm; ^1^H- and ^13^C-NMR data, Table 1 and Table 2; HRESIMS *m*/*z* 369.2031 [M + Na]^+^ (calcd for C_21_H_30_O_4_Na, 369.2036).

*Poloncosidin E* (**5**) colorless oil; ^1^H- and ^13^C-NMR data, Table 1 and Table 2; HRESIMS *m*/*z* 385.2372 [M + H]^+^ (calcd for C_24_H_33_O_4_, 385.2373), 407.2190 [M + Na]^+^ (calcd for C_24_H_32_O_4_Na, 407.2193).

*Poloncosidin F* (**6**) colorless oil; ${\left[\alpha \right]}_{D}^{25}$ = +30, c 0.10, MeOH; UV (MeOH) *λ*_max_ (log *ε*) 232 (2.95) nm, 301 (2.68) nm; ECD (1.98 mM, MeOH) *λ*_max_ (Δ*ε*) 210 (–3.36), 239 (–6.43), 300 (+3.40) nm; ^1^H- and ^13^C-NMR data, Table 1 and Table 2; HRESIMS *m*/*z* 405.2277 [M + H]^+^ (calcd for C_23_H_33_O_6_, 405.2272).

### 3.4. Fermentation, Extraction and Isolation

Conformational searches were carried out using molecular mechanics with the MM+ method and HyperChem 8.0 software. Afterward, the geometries were optimized at the gas-phase B3LYP/6-31G(d) level in Gaussian09 software to identify the energy-minimized conformers. Then, the optimized conformers were subjected to the calculations of ECD spectra using the TD-DFT at the BH&HLYP/TZVP level. Simultaneously, the solvent effects of the MeOH solution were evaluated at the same DFT level using the SCRF/PCM method [18].

### 3.5. ECD Calculations

Conformational searches were carried out via molecular mechanics with the MM+ method in HyperChem 8.0 software. Afterward, the geometries were optimized at the gas-phase B3LYP/6-31G(d) level in Gaussian09 software to identify the energy-minimized conformers. Then, the optimized conformers were subjected to calculations of the ECD spectra using the TDDFT at the BH&HLYP/TZVP level. Simultaneously, solvent effects of the MeOH solution were evaluated at the same DFT level using the SCRF/PCM method. [18]

### 3.6. Computational NMR Chemical Shift Calculations and DP4+ Analyses

All the theoretical calculations were performed using the Gaussian 09 program package. Conformational searches for possible isomers were conducted with molecular mechanics using the MMFF method with Macromodel software (Schrödinger, LLC) and the corresponding stable conformer, from which distributions higher than 2% were collected. Then, the B3LYP/6-31G(d) PCM level in DMSO was used to optimize the conformers. After that, the NMR shielding tensors of all optimized conformers were calculated using the DFT method at the mPW1PW91\6-31+G(d) PCM level on DMSO and then averaged based on Boltzmann’s distribution theory [16]. GIAO (gauge-independent atomic orbital) NMR chemical calculations were performed using an equation described previously [19]. Finally, the NMR chemical shifts and shielding tensors (^1^H and ^13^C) were analyzed and compared with the experimental chemical shifts using DP4+ probability [16].

### 3.7. Antibacterial Assay

The antimicrobial activities of the isolated compounds against the human and aquatic pathogenic bacteria, *Aeromonas hydrophilia* QDIO-1, *Edwardsiella ictarda* QDIO-9, *E. tarda* QDIO-2, *Escherichia coli* EMBLC-1, *Klebsiella pneumoniae* EMBLC-3, *S. aureus* (MASA) EMBLC-4, *Micrococcus luteus* QDIO-3, *Pseudomonas aeruginosa* QDIO-4, *Vibrio alginolyticus* QDIO-5, *V. anguillarum* QDIO-6, *V. harveyi* QDIO-7, *V. parahemolyticus* QDIO-8, and *V. vulnificus* QDIO-10, and the plant pathogenic fungi *Alternaria solani* QDAU-1, *Colletotrichum gloeosporioides* QDAU-2, *Fusarium graminearum* QDAU-4, *F. oxysporum* QDAU-8, *Gaeumannomyces graminis* QDAU-21, *Rhizoctonia cerealis* QDAU-20, and *Valsa mali* QDAU-16 were determined with a serial dilution technique using 96-well microtiter plates as in our previous report [14]. The human and aquatic pathogenic bacteria and plant pathogenic fungi were supplied by the Institute of Oceanology, Chinese Academy of Sciences.

## 4. Conclusions

In summary, six new verrucosidin derivatives (**1**–**6**) were identified from the deep sea cold-seep-derived *Penicillium polonicum* CS-252. Compounds **1**–**5** represent the first examples of verrucosidins with a 2,5-dihydrofuran ring. The structures of these compounds were determined using a combination of methods (NMR, MS, ECD calculations and DP4+ probability analysis).

All these compounds were tested for antimicrobial activities. Compounds **4** and **6** displayed broad-spectrum inhibitory activity against *Escherichia coli* EMBLC-1, *Klebsiella pneumoniae* EMBLC-3, *Pseudomonas aeruginosa* QDIO-4, *Vibrio alginolyticus* QDIO-5, and *V. parahemolyticus* QDIO-8 with MIC values ranging from 4 to 32 μg/mL. Compound **1** showed activity against *P. aeruginosa* QDIO-4, *V. alginolyticus* QDIO-5, and *V. parahemolyticus* QDIO-8 with MIC values of 8.0, 8.0, and 4.0 μg/mL, respectively. Moreover, compound **7** was bactericidal against *P. aeruginosa* QDIO-4 and *V. parahemolyticus* QDIO-8 (each with an MIC value of 8.0 μg/mL).

## Figures and Tables

**Figure 1 ijms-23-05567-f001:**
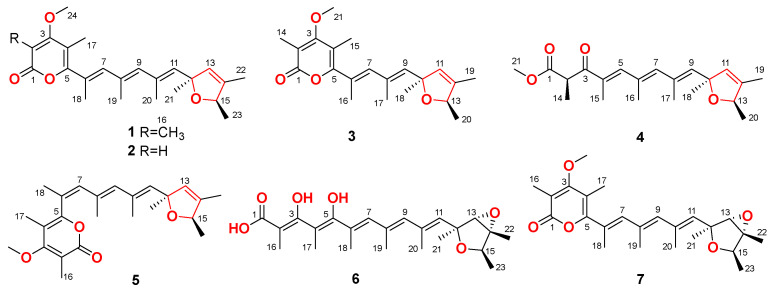
Chemical structures of compounds **1**–**7**.

**Figure 2 ijms-23-05567-f002:**
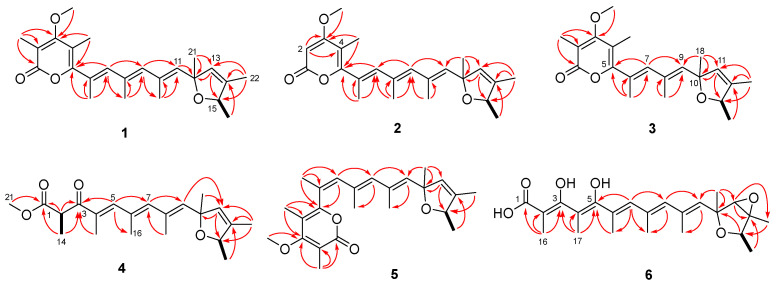
COSY (black bold lines) and key HMBC (red arrows) cross-peaks of compounds **1**–**6**.

**Figure 3 ijms-23-05567-f003:**
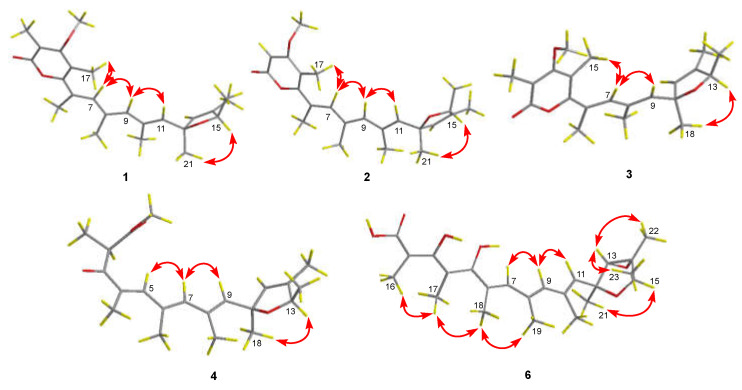
Key NOESY correlations of compounds **1**–**4** and **6**.

**Figure 4 ijms-23-05567-f004:**
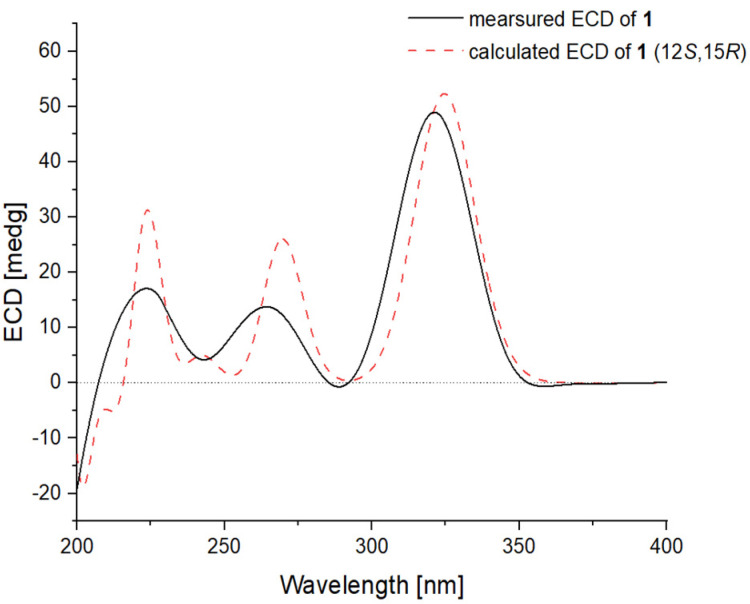
Experimental ECD spectrum of **1** (black line) compared with that calculated for (12*S*,15*R*)–isomer of **1** at BH&HLYP/TZVP level (red dashed line).

**Figure 5 ijms-23-05567-f005:**
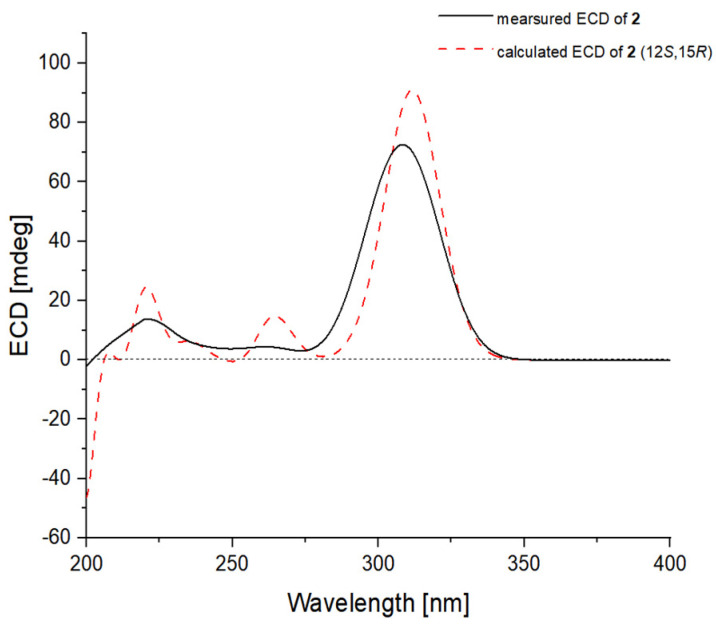
Experimental ECD spectrum of **2** (black line) comparing with that calculated for (12*S*,15*R*)–isomer of **2** at BH&HLYP/TZVP level (red dashed line).

**Figure 6 ijms-23-05567-f006:**
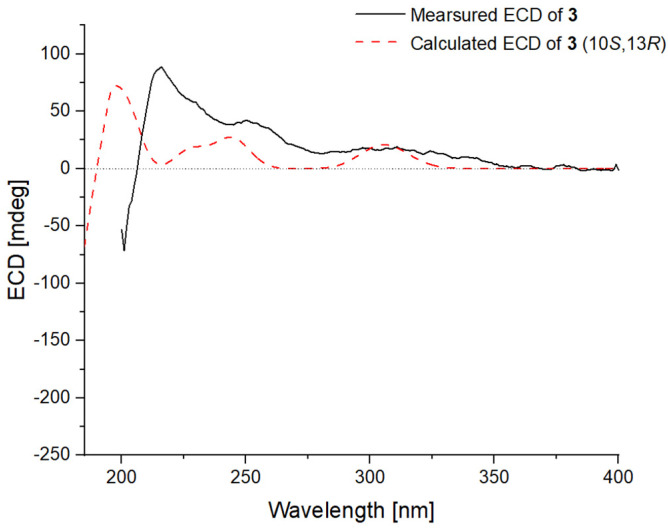
Experimental ECD spectrum of **3** (black line) comparing with that calculated for (10*S*,13*R*)–isomer of **3** at BH&HLYP/TZVP level (red dashed line).

**Figure 7 ijms-23-05567-f007:**
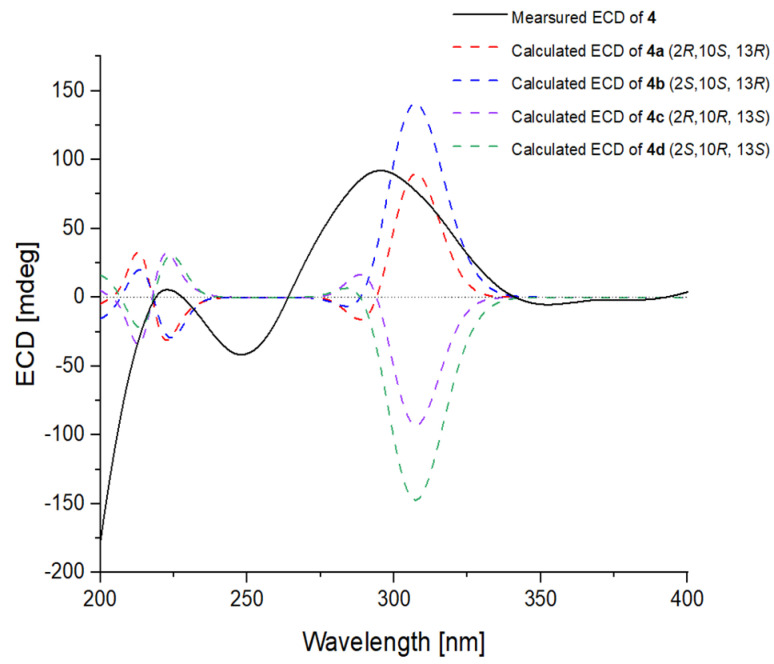
Experimental ECD spectrum of **4** (black line) compared with those calculated for **4a**–(2*R*,10*S*,13*R*), **4b**–(2*S*,10*S*,13*R*), **4c**–(2*R*,10*R*,13*S*), and **4d**–(2*S*,10*R*,13*S*) of **4** at BH&HLYP/TZVP level (red, blue, purple and green dashed lines, respectively).

**Figure 8 ijms-23-05567-f008:**
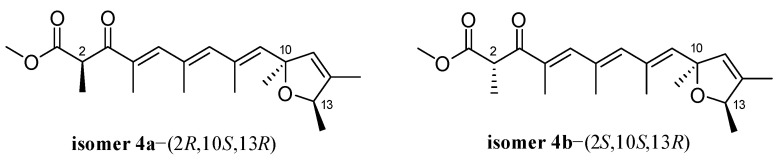
Two possible isomers of compound **4** for DP4+ probability analysis.

**Figure 9 ijms-23-05567-f009:**
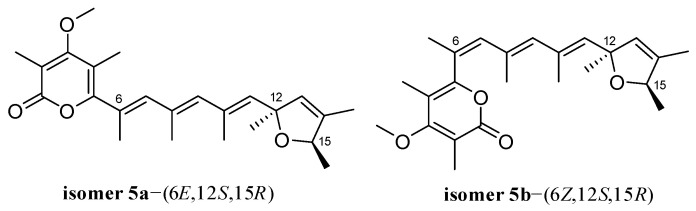
Two possible isomers of compound **5** for DP4+ probability analysis.

**Figure 10 ijms-23-05567-f010:**
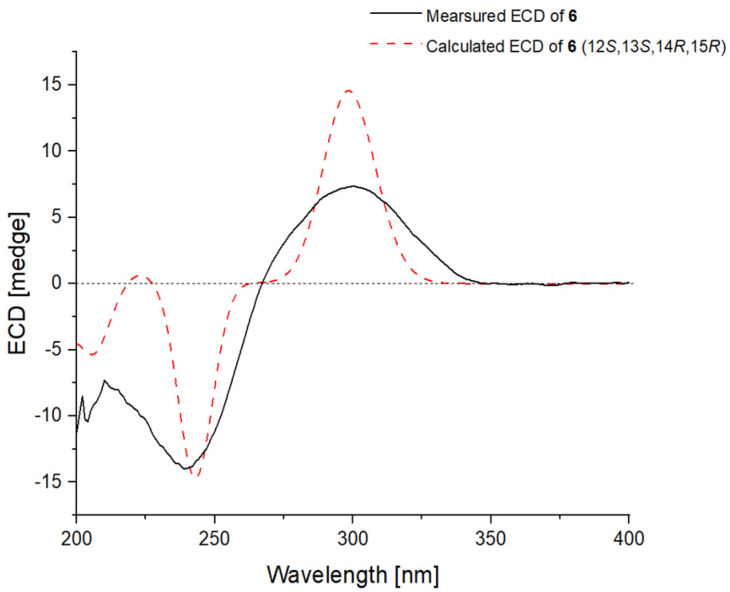
Experimental ECD spectrum of **6** (black line) compared with that calculated for (12*S*,13*S*,14*R*,15*R*)–isomer of **6** at BH&HLYP/TZVP level (red dashed line).

**Table 1 ijms-23-05567-t001:** ^1^H-NMR Data of Compounds **1**–**6** (500 MHz, DMSO-*d*_6_, *δ* in ppm, *J* in Hz).

No	1	2	3	4	5	6
2		5.61, s		4.52, q (7.0)		
5				7.19, s		
7	6.16, s	6.13, s	6.08, s	6.14, s	6.26, s	5.92, s
9	5.93, s	5.93, s	5.63, s	5.60, s	5.84, s	5.83, s
11	5.53, s	5.53, s	5.62, s	5.61, s	5.39, s	5.54, s
13	5.61, s	5.61, s	4.63, q (6.4)	4.63, q (6.4)	5.56, s	3.63, s
14			1.92, s	1.21, d (7.0)		
15	4.62, q (6.4)	4.61, q (6.4)	1.94, s	1.89, s	4.59, q (6.5)	4.00, q (6.7)
16	1.93, s		1.95, s	2.00, s	1.94, s	1.57, s
17	1.96, s	1.89, s	1.92, s	1.90, s	1.76, s	1.72, s
18	2.00, s	2.00, s	1.33, s	1.33, s	1.94, s	1.93, s
19	1.95, s	1.95, s	1.67, s	1.66, s	1.58, s	1.94, s
20	1.88, s	1.87, s	1.17, d (6.4)	1.16, d (6.4)	1.79, s	1.92, s
21	1.33, s	1.32, s	3.81, s	3.60, s	1.28, s	1.28, s
22	1.66, s	1.66, s			1.63, s	1.39, s
23	1.16, d (6.4)	1.16, d (6.4)			1.12, d (6.5)	1.11, d (6.7)
24	3.82, s	3.84, s			3.80, s	

**Table 2 ijms-23-05567-t002:** ^13^C-NMR Data of Compounds **1**–**6** (125 MHz, DMSO-*d*_6_, *δ* in ppm).

No	1	2	3	4	5	6
1	164.0, C	162.5, C	164.0, C	171.4, C	164.4, C	165.3, C
2	108.6, C	87.9, CH	108.8, C	45.6, CH	108.8, C	89.9, C
3	168.0, C	170.7, C	168.0, C	198.5, C	168.5, C	177.0, C
4	108.9, C	106.0, C	108.8, C	132.9, C	109.5, C	111.3, C
5	158.6, C	160.6, C	158.4, C	145.5, CH	155.9, C	155.4, C
6	126.3, C	126.3, C	126.3, C	131.8, C	125.6, C	132.3, C
7	139.1, CH	139.1, CH	138.9, CH	140.8, CH	133.0, CH	135.2, CH
8	131.3, C	131.2, C	130.1, C	130.6, C	131.3, C	129.7, C
9	137.1, CH	137.1, CH	138.3, CH	138.3, CH	137.3, CH	134.6, CH
10	130.7, C	130.7, C	87.3, C	87.3, C	130.6, C	134.4, C
11	137.2, CH	137.2, CH	127.1, CH	127.1, CH	137.7, CH	132.8, CH
12	87.3, C	87.3, C	138.4, C	138.4, C	87.3, C	79.4, C
13	127.2, CH	127.3, CH	81.6, CH	81.6, CH	127.2, CH	66.4, CH
14	138.2, C	138.3, C	9.9, CH_3_	14.3, CH_3_	138.3, C	66.9, C
15	81.6, CH	81.6, CH	11.6, CH_3_	13.0, CH_3_	81.6, CH	75.9, CH
16	9.9, CH_3_		15.9, CH_3_	17.8, CH_3_	10.0, CH_3_	9.9, CH_3_
17	11.7, CH_3_	10.8, CH_3_	17.4, CH_3_	17.6, CH_3_	10.5, CH_3_	12.3, CH_3_
18	16.2, CH_3_	16.2, CH_3_	27.6, CH_3_	27.6, CH_3_	22.7, CH_3_	16.7, CH_3_
19	17.9, CH_3_	17.9, CH_3_	11.8, CH_3_	11.8, CH_3_	15.7, CH_3_	18.4, CH_3_
20	17.8, CH_3_	17.9, CH_3_	20.5, CH_3_	20.5, CH_3_	17.7, CH_3_	18.3, CH_3_
21	27.6, CH_3_	27.7, CH_3_	60.2, CH_3_	51.9, CH_3_	27.6, CH_3_	21.9, CH_3_
22	11.6, CH_3_	11.8, CH_3_			11.8, CH_3_	13.4, CH_3_
23	20.5, CH_3_	20.5, CH_3_			20.5, CH_3_	18.6, CH_3_
24	60.2, CH_3_	56.7, CH_3_			60.3, CH_3_	

**Table 3 ijms-23-05567-t003:** Antimicrobial Activities of Compounds **1**–**4**, **6**, and **7** (MIC, μg/mL).

Strains	Compounds						
	1	2	3	4	6	7	Chloramphenicol *^b^*
*E. coli*	16	– *^a^*	–	4.0	32	32	1.0
*K. pneumoniae*	–	32	–	32	16	–	4
MRSA	–	–	–	16	–	–	4
*P. aeruginosa*	8.0	16	8.0	16	16	8.0	1.0
*V. alginolyticus*	8.0	–	16	8.0	8.0	32	0.5
*V. parahemolyticus*	4.0	8.0	–	16	16	8.0	1.0

*^a^* (–) = MIC > 32 μg/mL. *^b^* Chloramphenicol as positive control.

## Data Availability

Not applicable.

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
