# Peer review of "Verrucosidin Derivatives from the Deep Sea Cold-Seep-Derived Fungus Penicillium polonicum CS-252"

_ijms, 2022, doi:10.3390/ijms23105567_

Round 1

Reviewer 1 Report

This work is interesting and well presented.

Minor comments: 

1-Comments: Missing of Experimental ECD spectrum figure for compound 5

2- Comments: Figure 9. 6 at BH&HLYP/TZVP level (red dashed line)> should be Figure 10.

Reviewer 2 Report

This study has been well carried out and the results were found to be attracted and useful for the discovery of bioactivie natural products from marine fungi. I suggest the acceptance for publication of this work after the clarification of the questions which are as listed below.

  1. The title of this paper can be changed as compounds 4 and 6 do not possess the 2-pyrone ring which is one of the essential structure unit of verrucosidin-type compounds. Also,  compounds 1-3 and 5 are new compounds, but it might be not suitable to describe these compounds as unusual because they are simply the deoxygenated compounds from those discovered previously.
  2. The absolute configuration of 3 was determined by experimental and calculated ECD spectra, however, both spectra shown in Figure 6 are not well matched, in particular between 200 to 250 nm.
  3. It was pointed that the TDDFT-ECD calculations suggested that both conformers  4a-(2R,10S,13R) and 4b-(2S,10S,13R) were recommended for 4 (Figure 8), which were further clarified by the DP4+ probability analysis. Both proton and carbon data of the two possible isomers were calculated based on the DP4+ protocol and the results were analyzed with the experimental values . The statistical analysis indicated that the isomer  [4a-(2R,10S,13R)] was the equivalent structure with a probability of 99.91% (Figures 8 and 
    S42). The absolute configuration of 4 was thus assigned. However, in S42 it was found that the caululated proton and carbon data  were not provided, and there should be provided.
  4. It should be mentioned in the manuscript that compound 5 was isolated as a highly impured mixture, and also please discuss whether it is an artifact arisen from the photo-induced isomerization of compound 1.
  5. The structure of compound 6 is questionable. The NOESY spectrum was not well recorded so that I do not think that the NOE correlations mentioned in the manuscript which are important for accurate stereochemical assignment, such as between those olefinic methyls, could be found in the spectrum. Compound 6 might still be a compound with a pyron ring as the chemical shift of C-5 is quite close to C-5 shifts of compounds 1-3. The IR spectrum of 6 can solve the problem.
